# The Effectiveness of Agricultural Carbon Dioxide Removal using the University of Victoria Earth System Climate Model

Rebecca Evans[1] and H. Damon Matthews[1]

[1]Department of Geography, Planning & Environment, Concordia University, Montréal, Québec, Canada

**Correspondence:** Rebecca Evans (rebeccachloe.evans@concordia.ca)

**Abstract.** A growing body of evidence suggests that to achieve the temperature goals of the Paris Agreement, carbon dioxide removal (CDR) will likely be required in addition to massive carbon dioxide ($CO_2$) emissions reductions. Nature-based CDR, which includes a range of strategies to enhance carbon storage in natural and managed land reservoirs, such as agricultural lands, could play an important role in efforts to limit climate warming to well below 2°C above preindustrial levels. However, there remains a substantial knowledge gap on how the climate will respond to CDR when the removed carbon remains in the active carbon cycle. This study uses an intermediate complexity climate model to perform simulations of agricultural CDR via soil carbon sequestration at rates reflecting realistic costs under three future emissions scenarios. We found that plausible levels of agricultural CDR reduced $CO_2$ concentration by 5-19 ppm and global surface air temperature by 0.02-0.10°C by the end-of-century. This temperature decrease was non-linear with respect to cumulative removals, as the removed carbon remained part of the active carbon cycle, lessening the climate benefit than if it was removed permanently. In low emissions scenarios, a given amount of CDR was found to be more effective at reducing surface air temperature and less effective at reducing atmospheric $CO_2$, compared to high emissions scenarios. This was due to a proportionally larger impact of CDR on radiative balance at lower atmospheric $CO_2$, and reduced weakening of the carbon sinks at lower atmospheric $CO_2$. CDR was substantially more effective when implemented at a higher rate, as CDR makes a proportionally larger difference in a climate with lower cumulative air fraction of $CO_2$. Land and soil carbon responses were driven by the scenario-dependent balances between the impacts of CDR on primary productivity from $CO_2$ fertilization, and the impacts on soil respiration from increased soil carbon availability and global temperatures.

## 1 Introduction

To meet the goals of the Paris Agreement and limit warming to 2°C above the pre-industrial temperature, there must be a massive reduction in carbon dioxide ($CO_2$) emissions as well as the implementation of carbon dioxide removal (CDR) from the atmosphere (Rogeli et al., 2021; Huang and Zhai, 2021). Nature-based climate solutions (NbCS) are methods of CDR which enhance carbon storage beyond its natural level in natural and managed ecosystems, such as agricultural lands. Recent research has shown that soil carbon sequestration in croplands and pasturelands has the technical potential to sequester 0.38-9.34 $GtCO_2eq \ yr^{-1}$ between 2020 and 2050 (Roe et al., 2019). However, little is known about the transient climate response to removal when the sequestered carbon is not permanently removed but instead remains part of the active carbon

cycle. Furthermore, little is quantified about the impact agricultural CDR could have on the climate under different warming scenarios, particularly with respect to global temperature, carbon storage in soil and vegetation, and carbon fluxes.

Agriculture offers large-scale CDR in the near term using technologies that already exist, which collectively could be a substantial contribution to long-term global negative emissions in the future (Paustian et al., 2019). Enhancing soil carbon storage, through protecting existing carbon pools and rebuilding depleted ones, has the potential to provide over a quarter of the nature-based CDR required to keep climate warming below 2°C above preindustrial levels (Bossio et al., 2020). Regional empirical studies and modeling studies on specific components of agricultural CDR, such as biochar, agroforesty, nutrient management, and other management strategies, have shown promising results with respect to carbon sequestration and retention potential (ur Rehman et al., 2023; Tan and Kuebbing, 2023; Mason et al., 2023; T.M. et al., 2023; Wiltshire and Beckage, 2023; Chen et al., 2023; Lefebvre et al., 2024). For example, Chen et al. (2023) showed that increased soil carbon sequestration due to compost application in China alone could account for 0.31 $GtCO_2eq$ $yr^{-1}$ of removal by 2060. Lefebvre et al. (2024) found the application of biochar across the 746,000 ha of agricultural land in British Columbia removed 2.5 $MtCO_2eq$ during a simulated 20 year span.

Collectively the components of agricultural CDR have been estimated to have a mitigation potential of 3.0-8.5 $GtCO_2eq$ $yr^{-1}$ by 2050 accounting for constraints for food security and biodiversity (Griscom et al., 2017; Brack and King, 2021; Nabuurs and Coauthors, 2022; Smith et al., 2013; Paustian et al., 2019). This range is based on the estimated costs of implementation, with 3.0 $GtCO_2eq$ $yr^{-1}$ being possible for under 100 USD per $tCO_2eq$ $yr^{-1}$, and up to 8.5 $GtCO_2eq$ $yr^{-1}$ achievable for higher costs using frontier technologies (Griscom et al., 2017; Brack and King, 2021). The Intergovernmental Panel on Climate Change 6th Assessment Report estimates that a mitigation potential of 4.1±1.6 $GtCO_2eq$ $yr^{-1}$ by 2050 is likely possible for agricultural CDR (Nabuurs and Coauthors, 2022). Therefore, agricultural CDR of 3.0, 4.1, and 8.5 $GtCO_2eq$ $yr^{-1}$ by 2050 represent a range possible sequestration rates through low-, moderate-, and high- cost removal strategies and will encompass much of the analysis in this study.

Many previous modeling studies on the response of the climate to emissions have found that the relationship between cumulative emissions and temperature change is approximately linear and path-independent (Matthews et al., 2009; Allen et al., 2009). This linear relationship has also been found to be true in reverse for permanent removal, in which temperature decreases approximately linearly per unit of cumulative CDR, provided the climate system was in equilibrium before the CDR was applied (Zickfeld et al., 2016, 2021). In these studies, removal was achieved through permanent or geological removal, where the sequestered carbon is removed from the active carbon cycle entirely and thus no longer interacts with the atmosphere. However, there remains a research gap on how this relationship differs for nature-based CDR. In nature-based CDR, such as via agriculture, the sequestered carbon is only temporarily removed from the atmosphere and stored with residence times of months to decades. It thus continues to actively cycle with the atmosphere, suggesting that the relationship between cumulative removal and temperature will be non-linear as some of the removed carbon returns to the atmosphere. Furthermore, the capacity of natural systems such as soil to temporarily store carbon is affected by climate change itself (Keller et al., 2018; Seddon et al., 2020; Canadell et al., 2023; Nabuurs and Coauthors, 2022; Tao et al., 2023), suggesting that the relationship between cumulative removal and temperature will be path-dependent.

This study aims to explore these relationships using a similar approach to that of Matthews et al. (2022). Matthews et al. (2022) simulated reforestation as a NbCS using the same intermediate complexity climate model and emissions pathways as those used in this study. In their study, nature-based carbon sequestration was achieved through temporarily expanding then contracting forest area at varying rates in order to reach net zero $CO_2$ emissions by 2056. They explored the impact that temporary removal has on temperature when the sequestered carbon is slowly returned to the atmosphere after having reached its theoretical maximum mitigation potential. They found nature-based CDR was effective at delaying warming, and if implemented alongside aggressive emissions reductions it could decrease peak warming by up to 0.07°C. They further determined that long-term warming was only decreased if some of the enhanced carbon storage was permanent. This raises questions about the mathematical relationship between cumulative removal and temperature change when the carbon storage is not permanent.

While agricultural CDR has potential as a NbCS, there is a substantial knowledge gap how the climate system will respond to the removal, given that it is not permanent. We aim to assess the relationship between agricultural nature-based cumulative removal and the response of atmospheric temperature and $CO_2$ concentration, and how that relationship differs from that for permanent removal. Additionally, we will explore the impacts of agricultural CDR on carbon fluxes and storage in the land, soil, and vegetation carbon pools. To address these questions, we will perform simulations in an intermediate complexity model - the University of Victoria Earth System Climate Model - from the years 2000 to 2100, prescribing a flux of carbon from the atmosphere in to soil in agricultural lands across the globe. This flux will be applied at different rates, aimed to reflect the realistic costs of implementation of agricultural CDR. We expect the improved understanding of the responses of the land carbon pools and the atmosphere to agricultural CDR to contribute to a better understanding of nature-based solutions to climate change in general.

## 2 Methodology

To explore the impacts of agricultural CDR on the global climate, this study uses simulations designed to represent realistically-possible CDR from agriculture in an intermediate complexity global climate model. The simulations were performed using the University of Victoria Earth System Climate Model (UVic ESCM) version 2.10.

### 2.1 Model Description

The UVic ESCM is an intermediate-complexity global climate model, capable of simulating Earth's climate for long timescales at a lower computational cost, making it suitable for multi-century climate processes such as carbon cycle feedbacks (Weaver et al., 2001; Eby et al., 2009; Mengis et al., 2020). The UVic ESCM is one of the more complex of the intermediate complexity models, owing to its moderately high horizontal resolution in all model components (3.6°x1.8°), presence of sea-ice with rheology, fully coupled ocean model, and sediment processes. It has the same level of complexity of a general circulation model with the exception of the model atmosphere, which is heavily simplified to enhance computational efficiency, thus rendering it an intermediate complexity model. The current model version, 2.10, performs well with regards to changes in

historical temperature and carbon fluxes (Mengis et al., 2020). Published biases in the UVic ESCM version 2.10 include too large vegetation density in the tropics, too large changes of ocean heat content, and too low oxygen utilisation in the Southern Ocean (Mengis et al., 2020).

The atmospheric component of the UVic ESCM is a two-dimensional energy-moisture balance model using thermodynamics instead of dynamics, parameterising atmospheric heat and moisture transport with diffusion (Fanning and Weaver, 1996). Wind velocity is prescribed based on NCAR/NCEP reanalysis data for monthly climatology (Eby et al., 2013). Using the prescribed wind fields, moisture, momentum, and heat fluxes are calculated in the model. Transient wind velocities are calculated based on anomalous surface pressure, caused by anomalies in surface temperature relative to the pre-industrial state (Weaver et al., 2001). The model does not simulate clouds, but instead produces rain or snow when relative humidity reaches 85%.

The oceanic component of the UVic ESCM is Modular Ocean Model 2 (MOM2), a fully three-dimensional ocean general circulation model consisting of 19 vertical levels, varying in vertical resolution from 50 m near the surface, to 500 m at depth (Bitz et al., 2001). The sea-ice component is a dynamical and thermodynamical model that is coupled to the ocean model and atmosphere model.

The land component of the model contains an elaborate representation of the carbon cycle. The land component is made up of a surface model, which is a simplified version of the Met Office Surface Exchange Scheme (MOSES), coupled to the vegetation model Top-down Representation of Interactive Foliage and Flora Including Dynamics (TRIFFID) (Meissner et al., 2003). Carbon fluxes are calculated in the MOSES model, which then modifies the land, soil, and vegetation carbon pools (Matthews et al., 2004). TRIFFID simulates the soil carbon and coverage of five plant functional types: broadleaf tree, needleleaf tree, C3 grasses (cool season frost tolerant grasses), C4 grasses (warm season), and shrubs. In the TRIFFID model, agricultural crops are treated as C3 grasses. The PFTs space competition routine is based on the Lotka-Volterra equations (Cox, 2001; Meissner et al., 2003). In the UVic ESCM most recent update (2.10), one major improvement was to soil carbon and hydrology (Mengis et al., 2020).

## 2.2 Simulation Design

The UVic ESCM was spun up for 10,000 years with atmospheric $CO_2$ levels prescribed at 285 ppm corresponding to the year 1850. The model was then run from 1850-2020 using historical emissions, then run under three Shared Socioeconomic Pathway (SSP) marker scenarios from 2020-2100 using projected emissions (Riahi et al., 2017; Meinshausen et al., 2020).

The historical emissions and SSPs used here are shown in Figure 1. They describe potential pathways in which global societal and economic structure will change in the coming century and are used to derive corresponding greenhouse gas emissions based on policies. Under SSP1, future socioeconomic development would be highly sustainable, leading to net-negative $CO_2$ emissions by 2055 (Riahi et al., 2017). Under SSP2, future conditions are similar to those of today, with slow progress and regional rivalry inhibiting sustainable development. Under SSP5, socioeconomic development exploits fossil fuels, accelerating $CO_2$ emissions to over 120 $GtCO_2$ yr$^{-1}$ by 2100. For each SSP marker scenario, the radiative forcing by 2100 is 1.9, 4.5, and 8.5 W m$^{-2}$ respectively. SSPs 1-1.9, and 2-4.5 represent the most likely range of scenarios for global development. The data used here were taken from the International Institute for Applied System Analysis SSP database version 2.0 (Riahi et al., 2017;

Meinshausen et al., 2020), which compiles historical emissions inventories (Velders et al., 2015; van Marle et al., 2017; Hoesly et al., 2018; Gütschow et al., 2016; Carpenter et al., 2014; Miller et al., 2014), and the future emissions from the SSP1-1.9 marker scenario (van Vuuren et al., 2017), SSP2-4.5 marker scenario (Fricko et al., 2017), and SSP5-Baseline marker scenario (Kriegler et al., 2017).

For each of the three SSP's (1, 2, and 5), four simulations were performed in this study: one with no additional agricultural CDR (control), one with agricultural CDR that can be achieved for low costs (3.0 GtCO$_2$ yr$^{-1}$ globally by 2050), one with moderate agricultural CDR (4.1 GtCO$_2$ yr$^{-1}$ by 2050), and one with agricultural CDR that can be achieved for high costs (8.5 GtCO$_2$ yr$^{-1}$ by 2050). These will hereafter be referred to as no-, low-, moderate-, and high-removal. Thus there are a total of twelve simulations in this study.

The agricultural CDR was achieved by prescribing an atmosphere-to-soil carbon flux in agricultural areas. This flux was defined to be in addition to the existing model geochemical fluxes that affect soil carbon: gross primary productivity, soil respiration, and litter flux. Thus any responses of these three fluxes to CDR is a legitimate biogeochemical response and was not externally prescribed. The atmosphere-to-soil carbon flux is a simplified flux that was derived by summing the mitigation potentials of each component of agricultural CDR, such as biochar, nutrient management, etc. This choice was made on the basis that some of the components of the CDR cannot presently be modeled individually in the UVic ESCM, such as pyrogenic carbon storage from biochar and human activities in agriculture. The simple atmosphere-to-soil carbon flux varies according to the area of agricultural land in the grid cell. While some components of agricultural CDR are more effective in some regions than others, this was not incorporated in to the spatial variability of the prescribed flux in this study. This choice was made because many of the components of agricultural CDR have geographically sparse data available concerning their efficacy, and would require scientifically dubious interpolation.

The prescribed agricultural atmosphere-to-soil CDR flux was weighted by the fractional area of agriculture in the cell, which is shown in Figure 2. The agricultural area fraction was not prescribed to change after 2020. The global total of the flux was prescribed to be time varying, increasing linearly from 0.0 GtCO$_2$ yr$^{-1}$ at the year 2020 to 3.0, 4.1, or 8.5 GtCO$_2$ yr$^{-1}$ by 2050, after which the CDR was held constant as shown in Figure 2. At each model numerical integration step, at the computational stage when net atmosphere-soil carbon flux is calculated (which is the simple sum of net primary productivity, leaf litter flux, and soil respiration) an additional flux term was added to represent the spatiotemporally varying agricultural CDR atmosphere-to-soil flux. The magnitude of this flux was calculated based on the duration of the model time step and the annual flux for that model time step in Figure 2, and also weighted based on geographic location as outlined above.

## 3 Results and Discussion

### 3.1 CDR Impact on Atmospheric CO$_2$ Concentration and Temperature

Realistically-possible agricultural CDR was found in this study to have a tangible impact on CO$_2$ concentration and global surface air temperature (SAT) above the preindustrial value. As shown in Figure 3, by the end-of-century (EOC), in the low-removal scenarios, global SAT decreased by 0.02-0.04°C and CO$_2$ decreased by 5-7 ppm. Whereas high-removal resulted in

cooling between 0.06-0.1°C and $CO_2$ decline by 14-19 ppm. This shows that while the impact on global SAT is not enormous, the response of the climate to agricultural CDR is scenario dependent, so the same amount of removal in one scenario does not yield the same $CO_2$ decrease or temperature decrease as another scenario even though the simulations were initiated from the same transient state.

## 3.2 Change in Surface Air Temperature and $CO_2$ Concentration Per Unit of CDR

To explore the effectiveness of CDR, this study used an adaptation of the Transient Climate Response to Emissions known as the Transient Climate Response to Removals (TCRR) (Matthews et al., 2009; Zickfeld et al., 2021). TCRR is defined as the change in SAT over a given period (in this case 2020-2100) divided by the cumulative $CO_2$ removed in that time. The TCRR for this study is shown in Figure 4a, and the response of atmospheric $CO_2$ to cumulative removal is shown in Figure 4b.

This study found that the TCRR from agriculture is strongly non-linear, with the SAT decrease substantially slowing as
removal continues, and is also strongly dependent on the SSP scenario and rate of CDR (Figure 4a). For the higher emissions scenario scenario (SSP5), a given amount of CDR produced less of a temperature benefit than it did for the lower emissions scenarios (SSP1 and SSP2). For all scenarios, the CDR was less effective at reducing SAT when the CDR was implemented at a lower rate. For example, for SSP5, 50 GtC of CDR yields a temperature decrease of 0.2°C when implemented at the lowest rate, and 0.4°C for the highest rate.

The response in atmospheric $CO_2$ due to cumulative agricultural CDR was also found to be non-linear, with the CDR becoming less effective at decreasing $CO_2$ as removal continues (Figure 4b). The $CO_2$ benefit was also found to be weaker when CDR was implemented at lower rates. However, unlike for SAT, the $CO_2$ benefit from CDR was found to be higher in the high emission scenario than the lower emissions ones. Thus for any given amount of CDR, the $CO_2$ benefit from CDR is weaker and the SAT benefit is stronger in SSP1 than in SSP5.

### 3.2.1 Non-linearity of the TCRR (SAT) and $CO_2$

The deviation of TCRR from linearity is significant. Previous studies have shown that for geological CDR, in which carbon is permanently removed from the active carbon cycle, the TCRR is linear and only deviates from linearity when the initial climate state in which CDR is applied is not in equilibrium (Jones et al., 2016; Zickfeld et al., 2016, 2021). Furthermore, the TCRR in these studies was not scenario dependent, and instead only depended on the quantity of the cumulative removal. However, the
results shown here illustrate that for nature-based CDR, in which the carbon is not permanently removed but instead remains part of the active carbon cycle, the decline in SAT with CDR is non-linear and slows with increasing CDR.

For agricultural CDR and indeed nature-based CDR more generally, more carbon is being stored in natural systems, in this case soil, but this carbon remains part of the active carbon cycle. As a result, some of the removed carbon is returned to the atmosphere via soil respiration, meaning that per unit of CDR there is less of a cooling effect than if the carbon was removed
entirely. Thus with more CDR and more respiration, this effect saturates so CDR becomes less effective at reducing SAT because the carbon is more actively cycling. While the TCRR from gross-CDR is non-linear, it is possible that the TCRR from net-CDR is linear, although it was not possible to accurately quantify this for this study.

As for $CO_2$, the deviation from linearity occurs for the same reason, where the removed carbon remains in the active carbon cycle, continuing to respire back in to the atmosphere, so per unit of CDR there is less of a $CO_2$ decline than if the carbon was removed entirely. However, if the only factor affecting $CO_2$ and SAT was that carbon is not being permanently removed, we would expect the SAT benefit and $CO_2$ benefit to mirror each other for any given scenario. I.e. if the $CO_2$ benefit is weaker for SSP1 than SSP5, the SAT benefit would also be weaker in SSP1 than SSP5. This implies the importance of path-dependent additional effects such as impacts on radiative balance, and different responses the land carbon pools.

### 3.2.2 Path-Dependence of the TCRR (SAT) and $CO_2$

The scenario dependence of the TCRR and $CO_2$ benefit in this study is also a significant result, as it differs strongly from previous studies on geological CDR in which the TCRR is linear and path-independent. The SAT benefit of CDR deviates from linearity much more strongly for the higher emissions scenario (SSP5) than the lower emissions scenarios (SSPs 1 and 2). While for $CO_2$ the opposite is true, where the $CO_2$ benefit is closer to being linear for SSP5 than SSP1.

For agricultural CDR, the prescribed additional carbon flux in to the soil is partitioned by the UVic ESCM's land model in to additional carbon retained in the soil, additional carbon uptake by vegetation, and carbon returned to the atmosphere via soil respiration. The capacity of soil, vegetation, and the atmosphere to store carbon is strongly dependent on the climate, as is the interchange between those pools. Climate change directly and indirectly impacts the biogeochemical processes that determine the strength of the ocean and land carbon sinks. These impacts vary depending on the emissions scenario, thus the fraction of emitted $CO_2$ that remains in the atmosphere is scenario-dependent. This feedback can then amplify or weaken climate change through altering the global radiative balance.

High concentrations of $CO_2$ in the atmosphere cause cumulative ocean $CO_2$ uptake to be reduced due to the weakening of the buffering capacity of the ocean (Katavouta et al., 2018). The warming of the ocean also reduces its ability to dissolve $CO_2$, reducing ocean uptake further (Mathesius et al., 2015). Land carbon feedbacks are also strongly scenario dependent. Under high emissions scenarios, heat stress on vegetation, increased stomatal conductance and $CO_2$ fertilization, heat-induced increases in soil respiration, and permafrost carbon feedbacks together act to weaken the strength of the land sink relative to the amount of $CO_2$ emitted (Farquhar and Sharkey, 1982; King et al., 2004; Canadell et al., 2023; Jones et al., 2016). Together these mean that in a future with high emissions, the fraction of anthropogenic $CO_2$ that is absorbed by the land and ocean sinks will be significantly smaller than today, thus the cumulative airborne fraction of $CO_2$ is expected to be much larger than that under a low emissions scenario. This will drive a strengthening of the carbon cycle at higher emissions. To a first order, this is reversible, where negative emissions (removal) have a proportionally larger impact on atmospheric carbon storage in a high $CO_2$ climate as shown in Figure 5.

In Figure 5, the bars show the EOC difference between the amount of carbon stored in the CDR minus no-CDR scenarios for each of the land, ocean, and atmosphere pools. The percentages in each bar were computed as $100 \times |\Delta C_{stored-i}|/\sum C_{removed}$, where $|\Delta C_{stored}|$ is the absolute value at the EOC of the carbon stored in each pool, $i$, in the CDR minus the no-CDR scenario; and $\sum C_{removed}$ is the cumulative total of CDR by the EOC. For all SSP and CDR scenarios, the percentage of removed carbon that is retained in the land pool is around 1/3, which will be discussed further in Section 3.3. For all scenarios, there is an

increase in land carbon due to CDR, and a corresponding decrease in the amount of carbon stored in the ocean and atmosphere. For SSP1, the decrease in carbon stored in the ocean is around 10% of the total EOC CDR, while for the atmosphere the decrease is 20% of the CDR. For SSP5, the decrease in the ocean pool is proportionally much smaller at only 5%, while the decrease in the atmosphere carbon pool is much higher at 31%. So for a given amount of CDR by the EOC, only 20% of this will be removed from the atmosphere pool in SSP1, but 31% will be removed from the same pool in SSP5. This demonstrates that to a first order, a given amount of CDR will have a proportionally larger impact on the atmosphere carbon pool in a climate with high $CO_2$, even though the land carbon retention is approximately the same. For this reason, the lines for SSP1 in Figure 4b deviate more strongly from linearity than the lines for SSP5, as any given amount of CDR is less effective at inducing $CO_2$ drawdown in a lower emissions scenario.

Since the relationship between changes in atmospheric $CO_2$ and radiative forcing is logarithmic, at very high $CO_2$ concentrations such as in SSP5, a drop in atmospheric $CO_2$ due to CDR would have very little impact on radiative balance and therefore temperature (Matthews et al., 2009). For SSP1, atmospheric $CO_2$ concentration is lower, thus by the logarithmic relationship, CDR has a larger impact on radiative balance and therefore temperature. For this reason, the SAT benefit from CDR is higher for SSP1 than for SSP5.

### 3.2.3   CDR Rate-Dependence of the TCRR (SAT) and $CO_2$

For both SAT and $CO_2$, the response to cumulative CDR is weaker at lower rates of removal. For example for SSP1, 50 GtC of removal yields a temperature decrease of under 0.04°C when CDR is implemented at a low rate, and 0.05°C for a high rate. $CO_2$ concentration shows a similar pattern, with a decrease of 5 ppm after 50 GtC removed in SSP1, and 10 ppm for the same cumulative removal but at a higher rate. The rate of CDR dictates the year in which a given amount of cumulative removal is achieved, and therefore responses of SAT and $CO_2$ must be interpreted in the context of the background state of the climate at that time. For example, 50 GtC of cumulative removal is reached around the EOC for the low removal rate, but before 2055 for the high removal rate. The background states of the climate for all SSPs between 2055 and 2100 are very different. For SSP5, atmospheric $CO_2$ concentration at 2055 is around half of the value at 2100. Therefore, when CDR is implemented at a high rate, any given cumulative removal will be reached sooner and the background state of the climate will be cooler, so the CDR will have a larger impact on radiative balance and thus a larger impact on SAT. For this reason, for the same amount of cumulative removal, the CDR is more effective at reducing SAT if it is implemented at a higher rate as it has a stronger feedback on global radiative balance when atmospheric $CO_2$ is lower.

### 3.3   Land and Soil Carbon Pools

The results above imply that the entire land carbon cycle response to CDR is also strongly dependent on the emissions scenario and rate of removal. In this section we will focus on the land carbon response, and specifically the ability of soil to retain the carbon from the prescribed CDR. Since in these simulations the land surface is not prescribed to change, any impacts of CDR on land carbon should be solely a consequence of carbon cycle dynamics in a changing climate. The changes to the land, and specifically soil carbon pools are driven by the balance between increases in carbon due to direct uptake by plants and soil,

and decreases due to indirect impacts of climate change and CDR on vegetation and soil. The balance, and which processes dominate over one another, is scenario-dependent.

As shown in Figure 6a and b, CDR dramatically increases the storage of carbon in the land pool. These increases are dependent on both the SSP emissions scenario and CDR rate (Figure 6b). The changes in the land carbon pool are driven by the changes to vegetation carbon (Figures 6c and d) and soil carbon (Figures 6e and f). The impact of CDR on the land carbon

pool is dominated by the prescribed soil carbon flux.

Figures 6c and 6d illustrate the scenario-dependence of the response of vegetation carbon to CDR. In SSP5, vegetation carbon is largely unaffected by CDR since the CDR has a proportionally tiny impact on the massive atmospheric $CO_2$ concentration; thus the impact of CDR on $CO_2$ fertilization is negligible. In SSP1, the impact of CDR on vegetation carbon is dramatic and linear. Since atmospheric $CO_2$ is lower, CDR has a proportionally higher impact on the atmospheric $CO_2$ and thus more

strongly affects $CO_2$ fertilization, strongly decreasing vegetation carbon.

Figures 6e and 6f illustrate that the response of soil carbon to CDR is, unsurprisingly, more dependent on the rate of applied CDR than the emissions scenario. However, it is strongly non-linear, illustrating that as CDR continues less carbon is retained in the soil as it approaches saturation. Additionally, the soil carbon retention is slightly higher for the high emissions scenario than the lower one.

In theory we may have expected the lower emissions scenario to have better soil carbon retention, but the impact of CDR on $CO_2$ fertilization in SSP1 is strong, thereby weakening the carbon fluxes in to the soil via gross primary productivity (GPP) and leaf litter flux, which then reduces soil carbon retention. As shown in Figure 7, the impact of CDR on reducing GPP and leaf litter flux is much larger for SSP1 than SSPs 2 and 5. The impact also is substantially larger for higher rates of CDR because of the proportionally very large impact a higher rate of CDR has on $CO_2$ fertilization. This happens as a consequence of both

a larger annual CDR and a lower $CO_2$ background climate state. Soil respiration on the other hand only increases slightly in the early stages of CDR due to the flux of carbon in to the soil, then plateaus for increasing amounts of CDR. The initial increase occurs due to increased availability of carbon in the soil for microbial respiration, and the subsequent plateau occurs due to the balance of increased available soil carbon increasing respiration and decreased atmospheric temperatures reducing respiration. Therefore the two carbon fluxes in to the soil continually decline with increasing cumulative CDR, and the flux

out of the soil increases slightly then plateaus. Overall, the strong decrease in carbon flux in to the soil, and minimal increase in carbon flux out of the soil leads to slightly lower soil carbon retention in low emissions scenarios. In contrast for a higher emissions scenario, the impact of CDR on reducing GPP and leaf litter flux is substantially less due to its minimal impact on $CO_2$ fertilization. Soil respiration however increases almost linearly with CDR in SSP5, as soil respiration is not limited by a decrease in temperature like in SSP1. Thus GPP and leaf litter carbon fluxes in to the soil are high and minimally affected by

CDR, while there is more flux out of the soil from soil respiration. The net effect is that carbon fluxed in to soil via agricultural CDR is slightly better retained in the soil under high emissions scenarios than low emissions scenarios.

The percentage of carbon that is retained in the soil due to CDR is shown in Figure 8a. This was computed as the difference between soil carbon in the CDR minus no-CDR scenario divided by the prescribed carbon input in to the soil. As shown in Figure 8a, the percentage of removed carbon that remains in the soil declines strongly with increasing cumulative CDR, and

is strongly dependent on the rate of CDR. While for any given rate, slightly more soil carbon is retained in the soil for higher emissions scenarios, the more important factor appears to the rate at which the CDR is applied. In all scenarios, soil carbon retention reached around 35% by the EOC, meaning almost two thirds of the carbon fluxed in to soil through CDR cycled back in to the atmosphere. By the EOC, the percentage of soil carbon from CDR that was retained in soil was found to be strongly regionally varying and independent of the rate of CDR and scenario. Figures 8b-d show the regional pattern of the increase in soil carbon by the EOC in the CDR scenario minus the no-CDR scenario divided by the regionally varying cumulative carbon input. The spatial panels show the percentage soil retention for the low-removal scenario for SSPs 1, 2, and 5. The spatial pattern is identical for the moderate and high removal scenarios (not shown). The red box shows an example of an area where there is very little CDR applied, due to the presence of present day forests, but a large increase in soil carbon. This is likely due to the climate being overall more favourable due to CDR elsewhere, meaning the increases in soil carbon that would have happened in mid-latitude forests anyway, in the absence of CDR, was improved by the impact of CDR on the global climate even though CDR wasn't applied in that specific location. The blue box shows an area in which the soil carbon retained by the EOC is aligned with the global average. The yellow boxes show locations in which CDR was applied, but very little carbon was retained. This was because of strong soil respiration in these areas (not shown). This illustrates that the ability of global soils to retain any removed carbon in the soil is not spatially uniform, and is instead highly heterogeneous in space. Areas which are predicted to show an increase in stored soil carbon in the absence of CDR, as given in Figure 5.26 of Canadell et al. (2023), showed an even larger increase in soil carbon after CDR even if the CDR was not applied in those areas.

## 4    Uncertainty and Limitations

The results above are subject to uncertainty, related to uncertainties in the marker SSP scenarios, uncertainty in the theoretical potentials of the components of agricultural CDR, and limitations of the simulation design.

For each of SSP's 1, 2, and 5, there exists a group of simulations guided by the same paradigms as the marker scenarios used here but for which the derived emissions are different. These differences arise as a result of different assumptions and subjective interpretations that are required to quantify the narrative for each scenario: "global sustainability" (SSP1), "middle-of-the-road" (SSP2), and "fossil-fueled development" (SSP3). Elements of this include assumptions for energy and food demand, land-use, population growth, the extent of emissions mitigation, etc. (van Vuuren et al., 2017; Fricko et al., 2017; Kriegler et al., 2017). For each SSP, many simulations with the same paradigm can be performed using different Integrated Assessment Models, each of which have their own intrinsic calculations for investments in energy and resultant carbon emission mitigation and carbon taxing. Consequently, there are a myriad of sources of variability between different simulations within the same SSP scenario paradigm. The marker scenarios used here were chosen because they are very commonly used in the climate research community, thus facilitating comparison with the results of other studies. In this study, a more rigorous quantification of the impact of agricultural CDR on climate could be achieved by using additional emissions scenarios from each SSP. The choice of a single marker scenario per SSP is an acknowledged limitation of this study.

Agricultural CDR is itself composed of many constituent natural pathways, including but not limited to biochar, nutrient management, optimal intensity grazing, and conservation agriculture. Each of these components involve agricultural management techniques that can influence soil carbon storage including biochar retaining carbon on decadal scales, mechanical aeration affecting soil respiration, cover crop rotation aiding soil quality improvements, no-till farming enhancing short-term carbon retention, etc. These land management techniques are not currently able to be modeled in UVic ESCM, which is a limitation of this study.

Furthermore, the 95% confidence interval for mitigation potential for some of the constituent natural pathways is very large. This is due to a substantial range in empirical estimates of their mitigation potentials. For example, the mitigation potential of grazing legumes is 0.2 GtCO$_2$eq yr$^{-1}$ by 2030 with a confidence interval of 0.05-1.5 GtCO$_2$eq yr$^{-1}$ (Griscom et al., 2017; Brack and King, 2021). Other methods have much narrower confidence intervals (a range under 0.2 GtCO$_2$eq yr$^{-1}$) due to a wider availability of empirical estimates and expert elicitation, such as for nutrient management and improved rice cultivation. The 95% confidence interval for the collective components of agricultural CDR is 2.65-8.75 GtCO$_2$eq yr$^{-1}$ by 2030. Evidently, there is considerable uncertainty in the estimated mitigation potential of agricultural CDR by the mid-century. The rates of agricultural CDR applied in this study (3.0, 4.1, and 8.5 GtCO$_2$eq yr$^{-1}$ by 2050) were chosen based on cost estimates in (Brack and King, 2021). These rates encompass much of the range of the confidence interval for the collective components of agricultural CDR, and thus provide a reasonable but not perfect representation of the uncertainty in the theoretical mitigation potential.

Finally, there are notable uncertainties introduced by the nature of the simulation design. Since the UVic ESCM is not presently capable of modeling managed land practices, many of the components of agricultural CDR could not be modeled individually. Therefore, a simplified atmosphere-to-soil carbon flux was used which was aimed to represent the summed components of agricultural CDR as closely as possible. However, in practice, some natural pathways are implemented much more effectively in some places than others, with respect to carbon sequestration and retention. Examples include improved rice cultivation which is much more effective as a method of CDR in south Asia than everywhere else, and optimal intensity grazing which is much more effective in Europe and East Asia than northern and central Africa (based on Figure S2 in (Griscom et al., 2017)). Some managed land practices also limit soil carbon retention, such as tilling and other disturbances of carbon reservoirs. An atmosphere-to-soil flux that does not account for the spatial heterogeneity in CDR efficacy undoubtedly introduces some uncertainty in the results. This study was intended to show, to the first order, the climate responses which could theoretically occur given some rates of agricultural CDR.

Direct comparison of our results with previous studies is challenging due to the limited availability of studies of this kind on global nature-based CDR. The results presented here do compare well with an available previous study on temporary nature-based carbon removal. Matthews et al. (2022) modeled CDR via carbon storage in forests which were temporarily expanded in area then contracted under the same SSP1 and SSP2 marker scenarios that were used here. In their moderate removal scenario, a cumulative removal of 173 GtCO$_2$ (47 GtC) was achieved by 2056. This is very close to the cumulative removal achieved by 2056 in the high-removal scenario in this study (8.5 GtCO$_2$ yr$^{-1}$) as shown in Figure 2. Matthews et al. (2022) found that 47 GtC of cumulative removal by 2056 generated a decrease in atmospheric CO$_2$ concentration of around 10-15 ppm compared to

the no removal scenarios. In this study, as shown in Figure 4, 47 GtC of cumulative removal results in a decrease in atmospheric $CO_2$ concentration of around 9-10 ppm relative to the no CDR scenario. While a direct comparison of our study with Matthews et al. (2022) is not exactly appropriate due to the vastly different approaches of achieving the CDR, this suggests our results are generally in good agreement with a previous study on global nature-based removal.

Given the uncertainties associated with the intermediate complexity simulation design in this study, these results should primarily be taken as a first-order illustration of the theoretically possible climate responses to agricultural CDR and an explanation of how it differs mechanistically from permanent carbon sequestration.

## 5 Implications

This study offers insights into our understanding of the transient climate response removal when the sequestered carbon is still actively cycling with the atmosphere, as is the case in the vast majority of nature-based pathways. The responses of atmospheric $CO_2$ concentration and surface air temperature to cumulative carbon removal was non-linear. Removal became less effective at inducing climate benefits over time as the removal continued. In all of the emissions scenarios, the temperature and $CO_2$ responses to removal were considerably larger when CDR was implemented at the highest rate and under strong emissions reductions (SSP1). This implies that agriculture as a method of CDR is, to some extent, only meaningfully beneficial at mitigating climate change if enacted strongly and alongside massive emissions reductions. A CDR-induced cooling of 0.1°C by the end-of-century is much more helpful in mitigating the negative impacts of climate change if the net warming since 2000 is under 1.5°C (as is the case for SSP1) than if warming profoundly eclipses the 2.0°C threshold (as in SSPs 2 and 5).

These findings also have implications for the practical implementation of agricultural CDR. For agricultural CDR to be meaningfully effective, it should ideally be implemented at 8.5 $GtCO_2e$ $yr^{-1}$ by 2050. For this magnitude of annual CDR to be achieved, it would require substantial financial investment, in particular for biochar, trees in croplands, grazing - feed, and grazing - animal management (Brack and King, 2021). As with any method of CDR, the economic and practical considerations of implementing agricultural CDR are tremendously complicated. While these considerations are outside the scope of this study, a rigorous exploration of their feasibility would be an important focus of future studies.

## 6 Conclusions

This study uses simulations of agricultural carbon dioxide removal (CDR) at varying rates to explore the impact of nature-based CDR on the climate and the global land carbon pools. The simulations were performed using the University of Victoria Earth System Climate Model version 2.10. The agricultural CDR was achieved through a prescribed carbon flux from the atmosphere in to soil in agricultural areas. This was prescribed to be time varying, from 0.0 $GtCO_2$ $yr^{-1}$ in 2020, to 3.0, 4.1, or 8.5 $GtCO_2$ $yr^{-1}$ by 2050 based on estimates of low-, moderate-, and high- costs of implementation. These removals were performed under derived emissions from the shared socioeconomic pathways (SSPs) marker scenarios 1-1.9, 2-4.5, and 5-8.5.

This study yielded an important finding, that for agricultural CDR, and indeed nature-based CDR more generally, the response of $CO_2$ and surface air temperature to cumulative carbon removal is non-linear. Their responses were also dependent on the emissions scenario in which the CDR was implemented, and the rate at which the CDR was applied. We found that realistically-possible agricultural CDR was able to reduce $CO_2$ concentration by 5-19 ppm and global surface air temperature by 0.02-0.10°C by the end of century. The transient climate response to removal was non-linear, with CDR becoming less effective at reducing $CO_2$ and surface air temperature as cumulative removal increased. This is because the carbon is not permanently removed in nature-based CDR, but remains part of the active carbon cycle. Therefore for a given amount of CDR, some of the carbon removed returns to the atmosphere via soil respiration so the climate benefit is less than if the carbon had been removed entirely.

The response of $CO_2$ and surface air temperature to agricultural CDR strongly depended on the scenario in which it was implemented. In low emissions scenarios, CDR was less effective at reducing atmospheric $CO_2$ for a given amount of CDR than the same amount of CDR in a high emissions scenario. On the other hand, in low emissions scenarios CDR was more effective at reducing the surface air temperature than it was in a high emissions scenario. The larger temperature response in low emissions scenarios was due to the logarithmic nature of the relationship between changes in atmospheric $CO_2$ concentration and the impact on radiative balance, where at low atmospheric $CO_2$ concentrations, CDR has a proportionally larger impact on atmospheric $CO_2$ and therefore radiative balance. CDR was substantially more effective at reducing surface air temperature when it was implemented at a more rapid rate.

The impact of CDR on land and soil carbon was determined by the balance between increases in carbon due to uptake by plants and soil, and decreases due to indirect impacts of climate change, such as soil respiration. In low emissions scenarios, agricultural CDR induced a a sharp decline in gross primary productivity and leaf litter flux due to the proportionally higher impact on the $CO_2$ fertilization effect. In low emissions scenarios, CDR only caused a slight increase in soil respiration due to more soil carbon availability. The net result was slightly lower soil carbon retention than for the high emissions scenario. In the high emissions scenario, which primary productivity was largely unaffected by CDR due to the logarithmic relationship between atmospheric $CO_2$ changes and the $CO_2$ fertilization effect, and soil respiration substantially increased due to soil carbon availability and large increases in global temperature. Thus for low emissions scenarios, the decrease in primary productivity due to CDR is important dictating the proportion of removed carbon that is retained in the soil, but for high emissions scenarios the CDR more strongly affected soil respiration. The soil carbon was found to be retained at a higher fraction for longer if the CDR rate was higher.

Further study on this topic should explore the climate impacts from agricultural CDR where a portion of the carbon removed enters the inactive carbon cycle through biochar pyrogenic carbon capture, and indeed to individually model the components of agricultural CDR. Furthermore, the impacts of land management practices, such as tilling, on soil carbon retention should be explored in a model setting.

*Code and data availability.* Model output data and code has been uploaded to the Canadian Federal Research Data Repository. https://doi.
org/10.20383/103.0978

*Author contributions.* Rebecca Evans performed the simulations, created figures, and wrote the manuscript. H. Damon Matthews provided extensive scientific guidance on the simulation design and data interpretation.

*Competing interests.* The authors declare that no competing interests are present.

*Acknowledgements.* The authors thank the Matthews Climate Lab at Concordia University for the helpful discussion during weekly group
meetings. The authors acknowledge helpful discussions with Amy Luers at Microsoft that contributed to the framing of this study. This work was made possible due to funding support from Microsoft, Environment and Climate Change Canada, and the Mitacs Elevate Postdoctoral Fellowship Program. It also benefited from computational resources provided by the Digital Research Alliance of Canada's Cedar cluster. The authors would also like to thank the reviewers for their extremely helpful comments in relation to improving this publication.

**References**

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

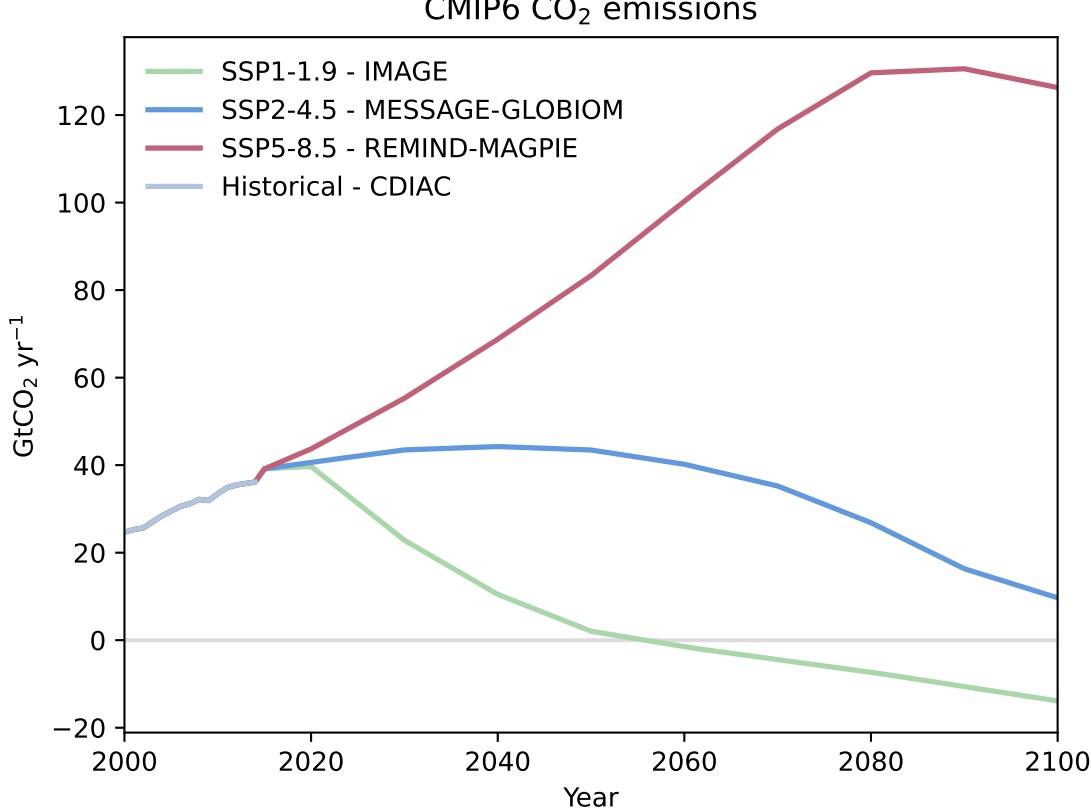

**Figure 1.** The three Shared Socioeconomic Pathways (SSPs) used in this study and historical $CO_2$ emissions. The SSP data is from 2015-2100 (Meinshausen et al., 2020). SSP1-1.9 is from the IMAGE integrated assessment model, SSP2-4.5 from MESSAGE-GLOBIOM, and SSP5-8.5 from REMIND-MAGPIE. Historical emissions were taken from the Carbon Dioxide Information Analysis Center.

**Figures**

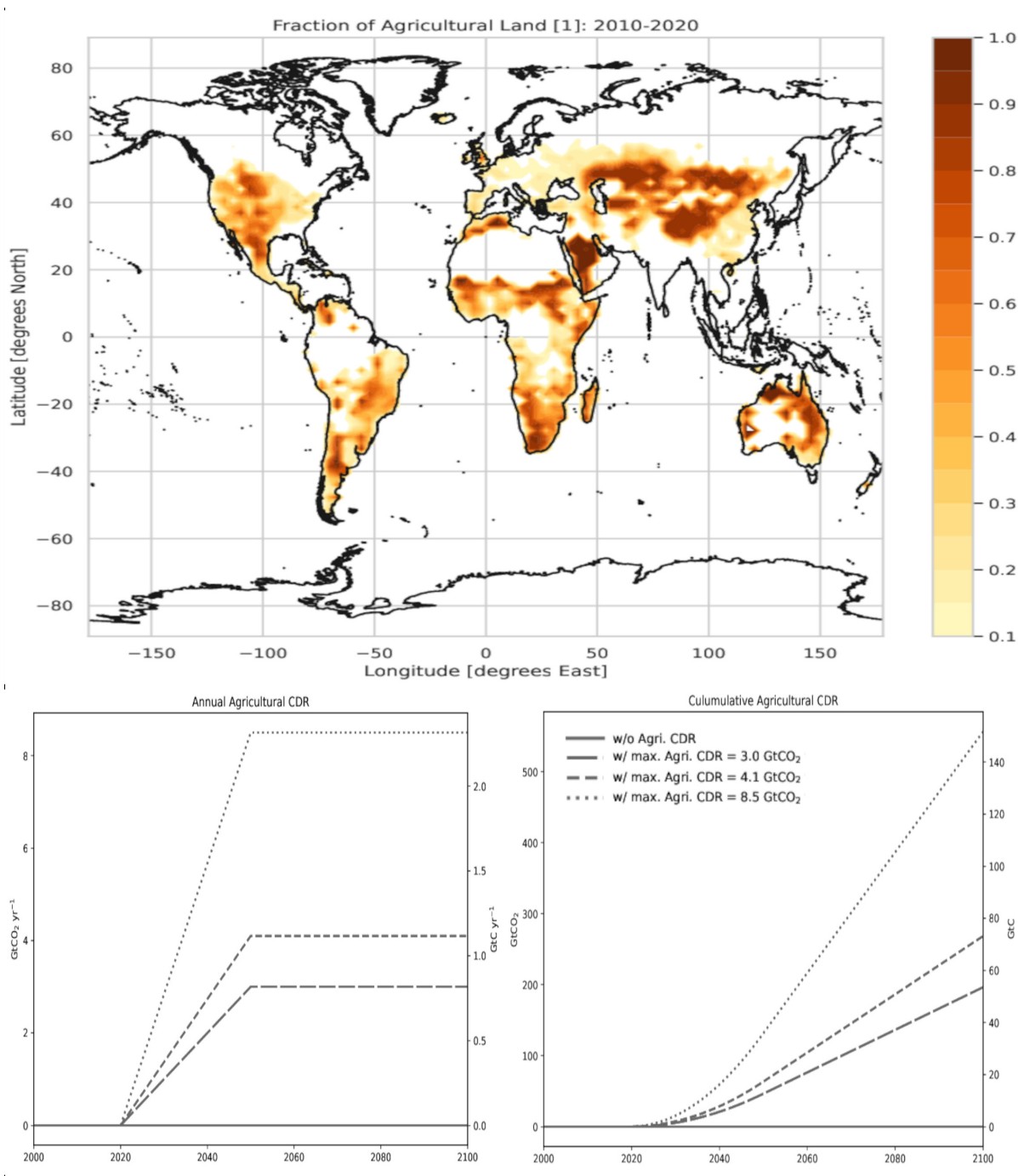

**Figure 2.** The upper plot shows the agricultural area fraction between 2010-2020, which was used to prescribe the locations of the CDR and amount of CDR per grid square. The lower plots show the annual and cumulative prescribed global agricultural CDR for the low removal (3.0 $GtCO_2$ $yr^{-1}$ by 2050), moderate removal (4.5 $GtCO_2$ $yr^{-1}$), and high removal scenarios (8.5 $GtCO_2$ $yr^{-1}$). Cumulative total removal by 2100 is 196.5 $GtCO_2$, 268.6 $GtCO_2$, and 556.8 $GtCO_2$ in the low-, moderate-, and high-removal scenarios respectively. The long-dashed, short-dashed, and dotted lines will hereafter be used to represent the low-, moderate-, and high-removal scenarios respectively.

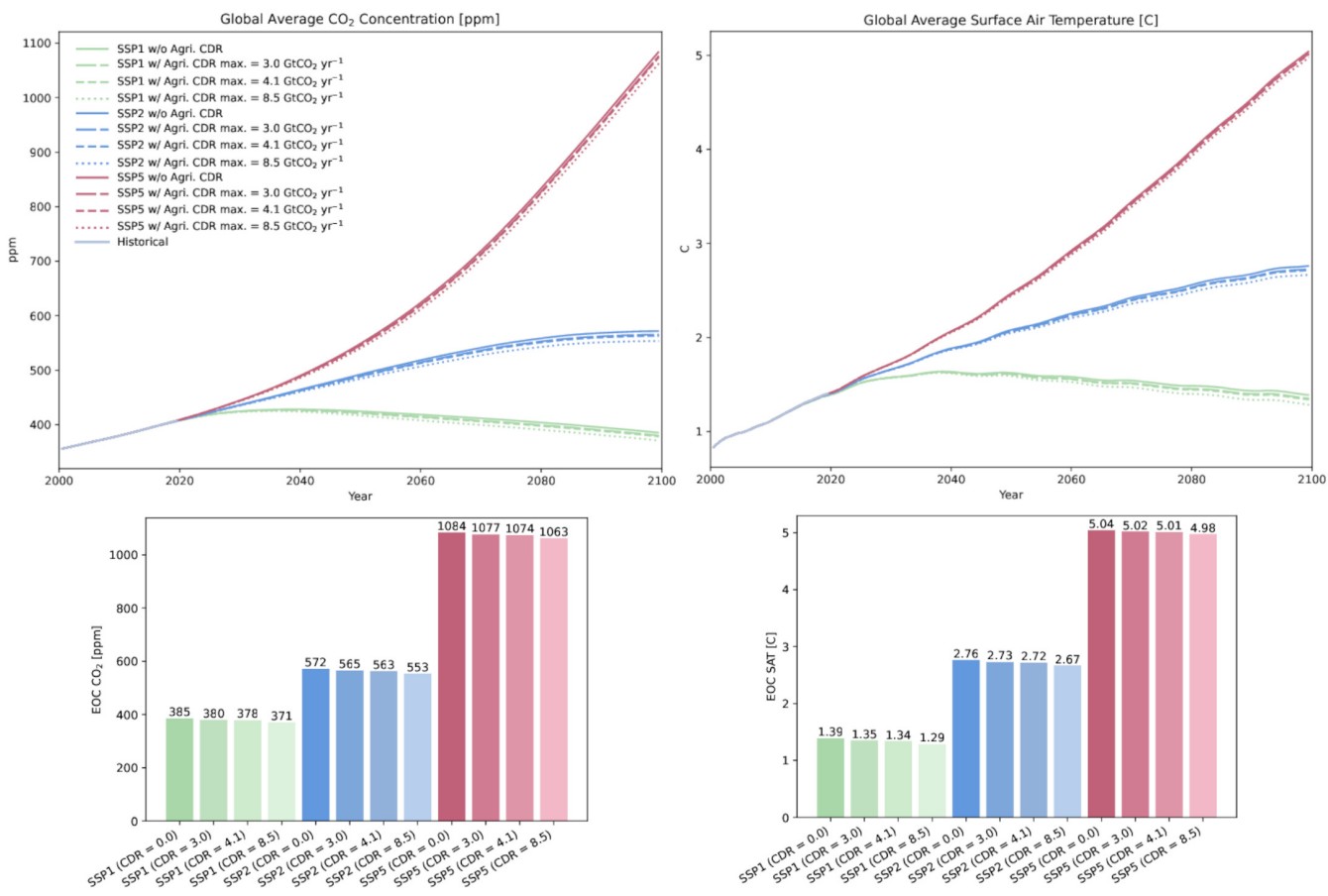

**Figure 3.** The upper line plots show the $CO_2$ concentration (left) and surface air temperature (right) with time. The lower bar charts show the End-of-century (EOC) $CO_2$ concentrations and surface air temperature in each simulation.

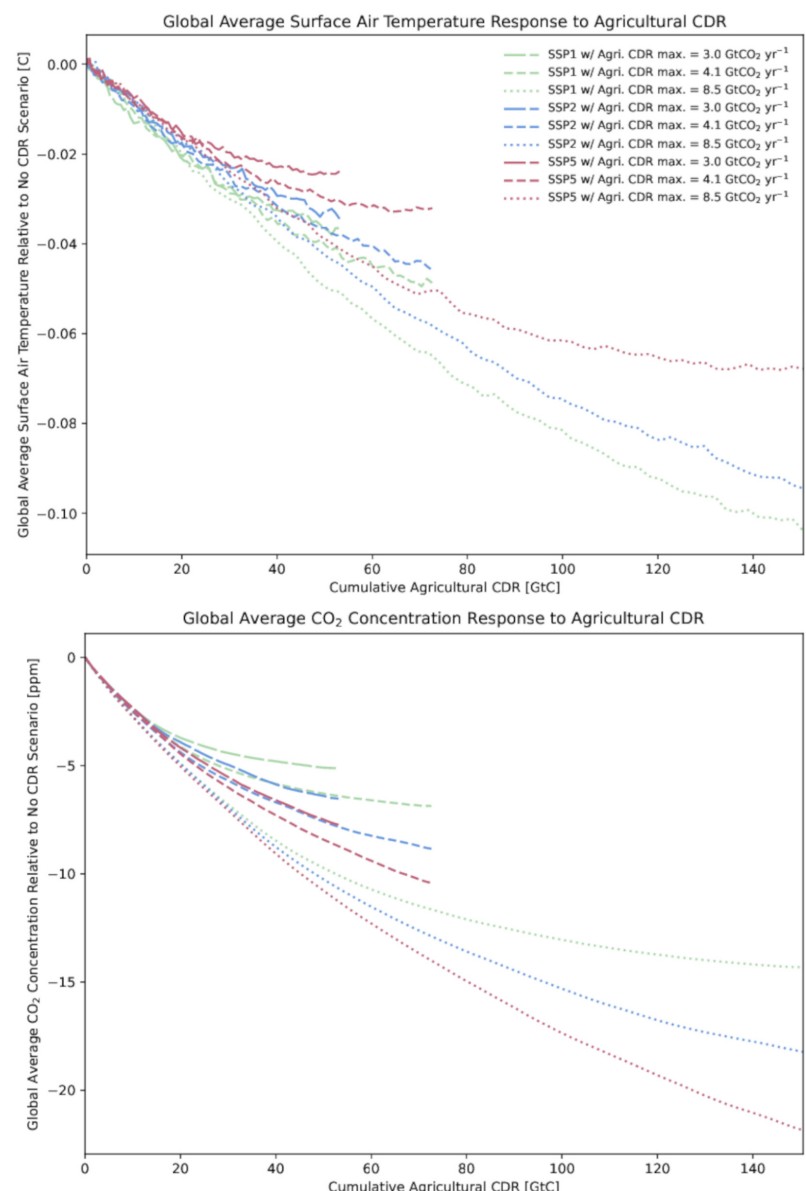

**Figure 4.** Transient Climate Response to Removal Plots. The uppermost plot (a) shows the global average surface air temperature response (CDR scenario minus no-CDR scenario) to cumulative removal. The bottom plot (b) shows the global $CO_2$ concentration response to cumulative removal.

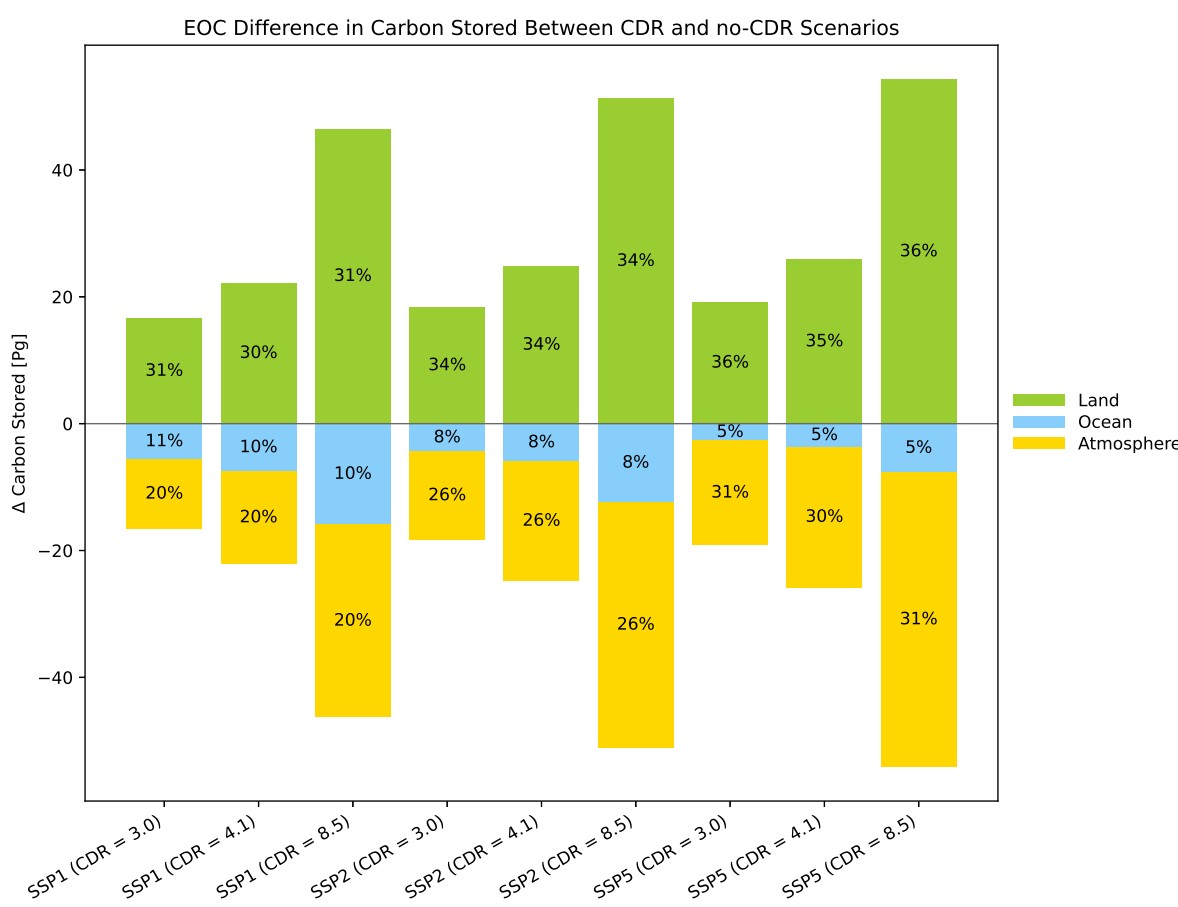

**Figure 5.** The difference at the end-of-century between the amount of carbon stored in the CDR minus no-CDR scenario for each of the land, ocean, and atmosphere pools. The percentages in each bar show the absolute value of the proportion of removed carbon retained in each pool.

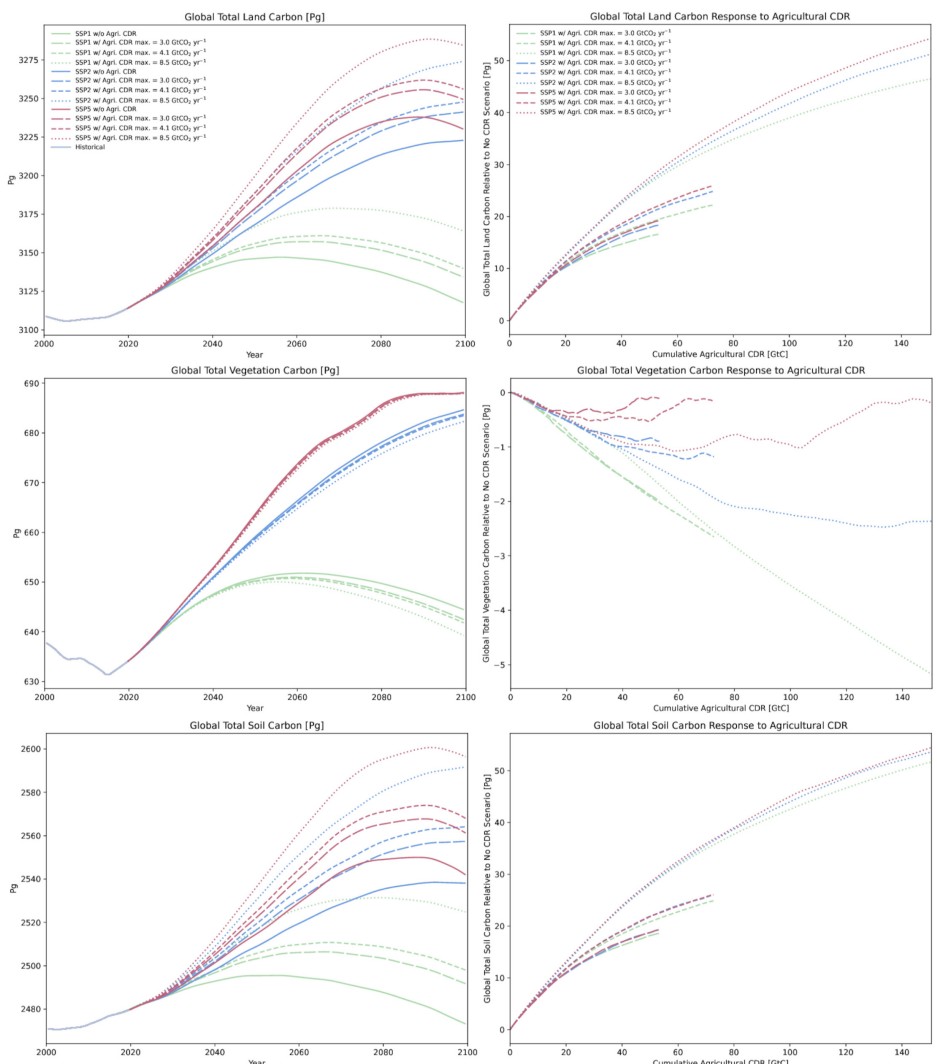

**Figure 6.** The global total land carbon pool (a, b) and its soil (c, d) and vegetation components (e, f). The left column shows the carbon storage totals with time. The right column shows the difference in carbon storage between the CDR and no-CDR scenarios against cumulative removal.

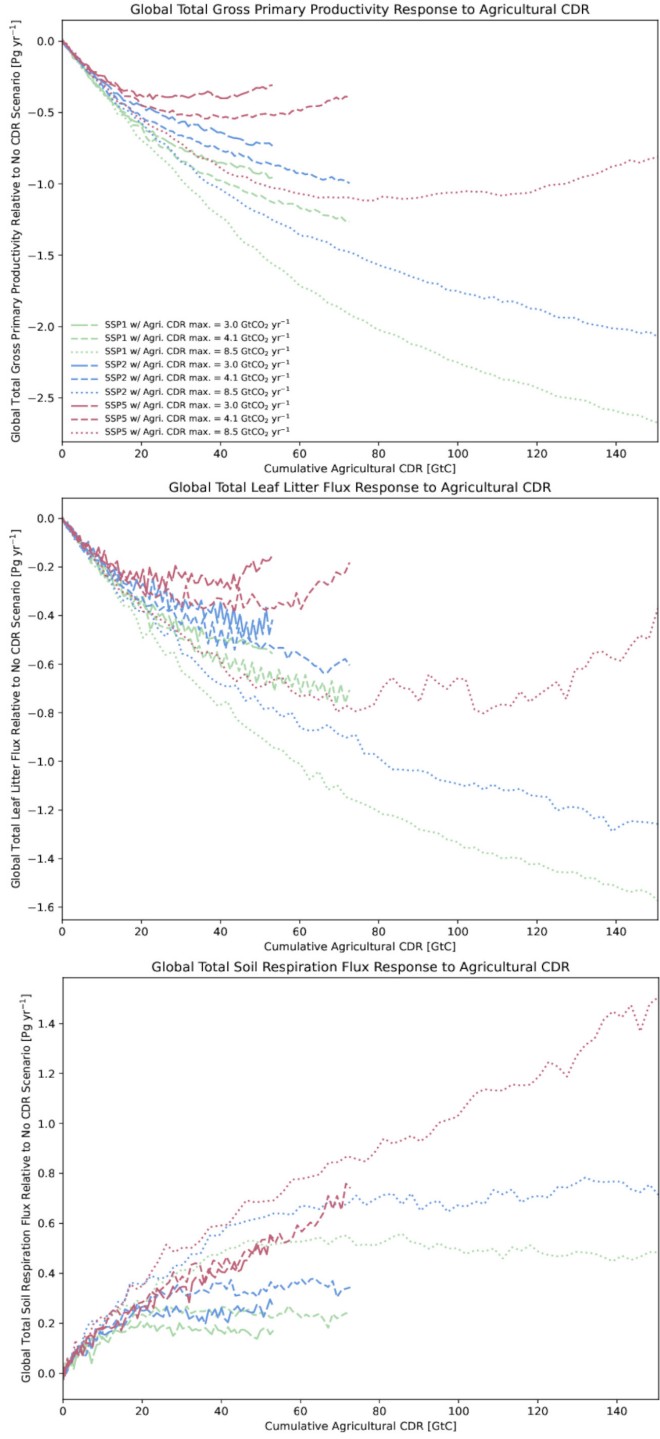

**Figure 7.** The components of soil carbon flux. The plots show the fluxes in the CDR scenarios minus the no-CDR scenarios against cumulative removal for (a) gross primary productivity (GPP), (b) soil respiration, and (c) leaf litter.

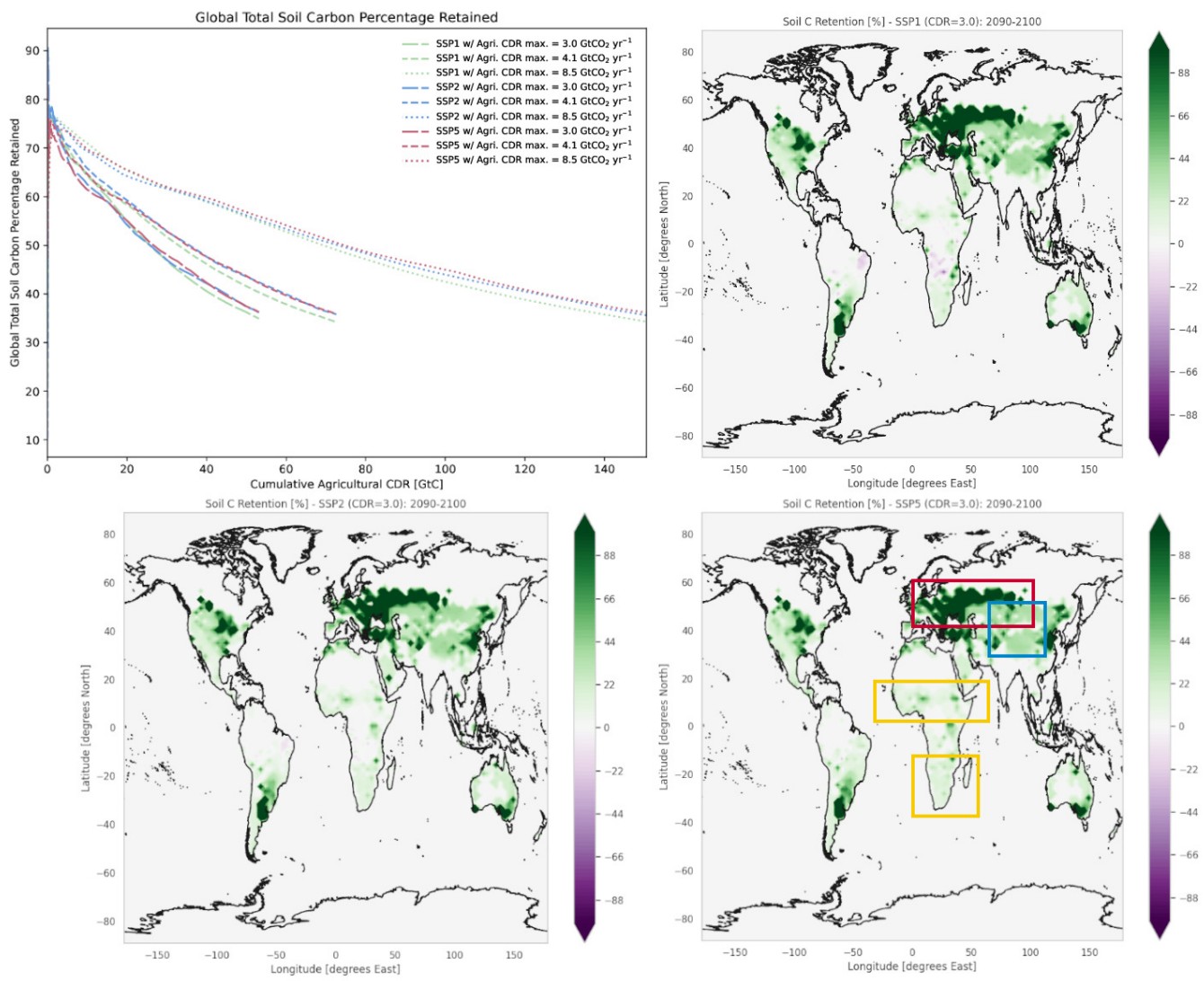

**Figure 8.** The percentage of carbon retained in the soil as (a) a function of time, and (b-d) a function of space averaged for the period 2090-2100 for SSP's 1, 2, and 5 for the low removal scenario. The spatial pattern of the plots in (b-d) is identical to that for the moderate- and high- removal scenarios (not shown). The red box shows an example of a region where there is little applied CDR, but very high C retention in soil; the blue box is an example of a region where the percentage of soil carbon retained is around the global average of 30%; the yellow boxes show examples of regions where CDR is prescribed to be strong but little soil C is retained. The regional variability of CDR can be found in Figure 2.