# Peer review of "The Effectiveness of Agricultural Carbon Dioxide Removal using the University of Victoria Earth System Climate Model"

_EGUsphere, 2024_

## Author Response (AR1)

**Reviewer 1:**

(Original comments shown in black; Responses shown in blue)

Overall comments:

This paper uses the UVic ESCM earth system model to evaluate and assess the effectiveness and impacts of agricultural carbon dioxide removal. Carbon dioxide removal is an important issue in the era of climate change, and nature based climate solutions are potential pathways to partially address the climate crisis. Therefore, the issue that the paper is trying to address is relevant to the broad geoscience community. The results and analysis on the model simulation are comprehensive and thorough, and several insights are highlighted based on the results.

However, there are some major issues of this paper, including language and writing, methodological rigor, and the lack of discussion on the limitations. I encourage the authors to address these issues and improve the paper before publishing.

Major comments:

1) The writing of this paper needs to be improved. Although the topic of this paper is of broad interest to the geoscience community, the way it is presented lacks clarity, especially the introduction section. In the introduction, the authors only mention the research gap in the last paragraph, which is confusing and unconventional for an introduction in a research paper. Instead, I suggest the authors restructure the introduction section to follow a more clear logic, for example, general topic – research gap – how this paper addresses the gap.

    a) The introduction has been completely rewritten to be more clear on the research gap and goals of the paper from the outset. Thank you for the feedback, we think it is in much better shape now.

2) There should be more explanation and discussions on how agricultural CDR is implemented. The paper only says that the agricultural CDR was achieved by prescribing an atmosphere-to-soil carbon flux in agricultural areas, and any changes in natural carbon storage and fluxes occurring will be considered as feedback effects. While I believe may be reasonable to do so, a more thorough explanation on how each type of CDR is implemented in real life (e.g., nutrient management, reforestations, etc.), and if it can be more accurately incorporated into the model should be provided. If there are more accurate ways of modeling them, the authors should adopt them. Also, even if the simplification is reasonable, the implications of this simplification should also be discussed. For example, will there be any constraint that the CDR and the feedback jointly have to satisfy?

    a) With the current UVic ESCM model setup, it is not currently possible to model some of the components of agricultural CDR (biochar, nutrient management etc.). This is because there is not specific treatment of managed land practices, and pyrogenic carbon storage due to biochar creation. A simplified flux was therefore used to represent these in the model without modeling them specifically. An explanation of this has been added to Section 2.2 (Simulation Design) on lines 136-154. Discussion of the associated uncertainty and limitations has been added to a new section (Section 4 - Uncertainty and Limitations). The discussion can be found on lines 327-354.

3) While the simulation results are comprehensive and thorough, the paper does not provide any uncertainty quantification, nor does it discuss any implications about uncertainty. I suggest the authors use some commonly used uncertainty quantification metrics for the results. If the uncertainty quantification is too challenging given the context or too cumbersome for this paper, possible sources of uncertainty and its implications should be at least discussed.

   a) A detailed description of the possible sources of uncertainty has been added to a new section (Section 4 - Uncertainty and Limitations). Given the at times large uncertainties associated with each SSP marker scenario, each component of agricultural CDR, and limited data available for some of those components, it was deemed too cumbersome to explicitly quantify the uncertainty. In lieu of that a detailed discussion of the implications of the uncertainty was added to that section.

4) I suggest the authors add a dedicated session to discuss the implications.

   a) We have added a new section on implications as suggested (Section 5)

**Reviewer 2:**

(Original comments shown in black; Responses shown in blue)

The manuscript investigates the potential for agricultural carbon dioxide removal (CDR) to mitigate climate change by simulating soil carbon sequestration using the University of Victoria Earth System Climate Model (UVic ESCM).

However, the paper's simplified treatment of agricultural CDR presents a key limitation by applying a uniform atmosphere-to-soil carbon flux across agricultural areas. This generalization does not account for the significant variability in the efficacy of different agricultural management practices, which critically affect carbon retention and sequestration longevity. Therefore, it's difficult to judge the robustness and uncertainty level of the assessed numbers. A comparison between the current study and previous assessment would be appreciated.

This is an important point to note and additional detail to explain this choice has been added to both the methodology (lines 136-154) and its implications discussed in a new section (Uncertainty and Limitations - Section 4).

The practical reason for which a uniform flux was chosen that does not account for spatial heterogeneity in efficacy was that the spatial data available for the efficacy of each component of agricultural CDR is very sparse. Many countries are missing data altogether, and others have only one data point available. For that reason, it was deemed to be too scientifically dubious to prescribe a spatially varying efficacy for this round of review as it would involve sparse data and a lot of interpolation. Hopefully in the future with more available data this will be possible.

Missed Mechanistic Detail:

Treating all agricultural CDR methods as a uniform flux from the atmosphere to the soil ignores the inherent complexities of soil carbon turnover. Different agricultural techniques impact microbial activity, soil structure, and decomposition rates differently. For example, practices that increase soil aeration (like tilling) could accelerate carbon release through respiration, undermining long-term sequestration.

> This is an acknowledged shortcoming of this study, owing to the lack of currently available modules in the UVic ESCM for managed land practices, and pyrogenic carbon sequestration through biochar. This shortcoming has been added to Section 4 (Uncertainty and Limitation) on lines 327-354.

> Specific mention of biochar, no till farming, cover crops, and aeration has been added to the same section, addressing the specific comments below.

> - Biochar: This method results in long-term carbon sequestration, as biochar can persist in soils for hundreds to thousands of years. Its resistance to decomposition means that biochar acts as a more permanent carbon sink.
>
> - No-Till Farming: While this method reduces soil disturbance and thereby helps increase soil organic carbon in the short term, its long-term efficacy may be more variable. The retained carbon could be vulnerable to re-release through natural processes like soil respiration.
>
> - Cover Cropping and Crop Rotations: These practices can increase soil organic matter, but the degree of sequestration depends heavily on crop types, climate, and soil properties. They generally offer more temporary sequestration benefits compared to biochar.

Limited Exploration of Economic and Practical Feasibility: While the authors mention the costs associated with low, moderate, and high CDR scenarios, there is no rigorous exploration of how economically feasible these interventions would be, or how they could be integrated into current agricultural systems without significant trade-offs.

> While the practical implementation of agricultural CDR is mostly outside the scope of this study, since it requires political/economic/management/social expertise that we simply do not possess, it is certainly worth mentioning. We have added a brief discussion in to a new section (Section 5 - Implications) on lines 379-384.

Another significant limitation of the paper lies in its failure to rigorously address uncertainties inherent in the Shared Socioeconomic Pathways (SSPs) used in the simulations. These pathways—SSP1-1.9, SSP2-4.5, and SSP5-8.5—represent different global development and emissions scenarios, but the uncertainties associated with these scenarios are not thoroughly explored.

> A discussion of the uncertainties associated the SSPs was added to the new Section 4 (Uncertainty and Limitations) on lines 315-327.

Overall, the manuscript is in good shape but could benefit from further refinement.

> Thank you!